# A dual-plane fluoroscope to track joint kinematics during dynamic daily activities

**Albert Planta**[ID]*, **Raphael Surbeck**[ID], **William R. Taylor**[ID], **Stefan Plüss**,
**Florian Vogl**, **Pascal Schütz**

Laboratory for Movement Biomechanics, ETH Zurich, Zurich, Switzerland

* albert.planta@hest.ethz.ch

## Abstract

Accurate measurement of joint kinematics is a key requirement for understanding injury mechanisms, evaluate rehabilitation techniques, and improve the implant designs and techniques used in total knee arthroplasty (TKA). Fluoroscopy is an experimental technique to directly measure joint kinematics without being affected by soft-tissue artefacts. However, because of its limited field-of-view (FOV), stationary fluoroscopy can only measure small parts of more dynamic/progressive movements, such as walking. This manuscript presents a new generation of moving fluoroscope: The tracking dual-plane fluoroscope (tDPF) combines optical tracking with a bi-planar X-ray system on mechanically independent source and intensifier carriages on rails and model-predictive-control to measure the kinematics of the tibio-femoral joint *in vivo* during dynamic activities, such as level walking and stair ascent/descent, at all gait speeds. In this proof-of-concept study, the tDPF tracked the knees of 16 young and healthy subjects during complete, consecutive gait cycles of level walking, ramp ascent, ramp descent, stair ascent, and stair descent at self-selected gait speeds. For all gait speeds (average and standard deviation: $1.34 \pm 0.14\,\mathrm{m\,s^{-1}}$ during level walking), tracking performance for each activity was excellent and the knee centre stayed within both simulated image intensifiers' FOVs for $> 99\%$ of frames (no X-ray images were captured in this study). The tDPF is the first dual-plane fluoroscope to track the knee joint during entire cycles of stair and ramp ascent at self-selected gait speeds for young and healthy subjects. Notably, our device does not require any pre-recording of movement patterns—by using real-time position estimates of the tracked joint and tracking each trial independently, even the challenging measurements of tasks with high variability between trials become possible.

## Introduction

The knee is one of the primary joints of the human body, and is involved in nearly every activity of daily living, such as gait, walking up stairs, or using a chair. Consequently, any injury or pathology affecting the knee severely reduces the quality of life of the affected per-

**Data availability statement:** All trackingcapability files are available in the polybox: https://polybox.ethz.ch/index.php/s/YOL1U9zVFU0MIdO.

**Funding:** The development of the tracking dual-plane fluoroscope was funded by Innosuisse (50579.1 IP-LS, Bern, Switzerland) with the Zimmer GmbH (Winterthur, Switzerland) acting as the implementation partner, and additionally sponsored by the Schulthess Fonds Maxistiftung (Zurich, Switzerland). The funders and sponsor did not play any role in the study design, data collection and analysis, decision to publish, or preparation of the manuscript.

**Competing interests:** The authors have declared that no competing interests exist.

son. Depending on the pathogenesis, rehabilitation treatments or orthopaedic interventions, such as total knee arthroplasty (TKA), can relieve patient pain and restore joint functionality. Ideally, these treatments would restore the functionality of the knee to its healthy state, including internal forces and joint kinematics. Thus, a detailed understanding of the kinematics of the healthy, affected, and restored knee are key requirements to evaluate and improve implant designs, validate knee models, and refine surgical or rehabilitation techniques. Compared to methods that measure the skin-surface (e.g. motion capture), videofluoroscopy directly measures the bones' kinematics and has been the technique of choice to investigate knee mechanics during level gait, sitting tasks, ramp descent, and even during a golf swing [1–7,21]. However, videofluoroscopy suffers from the drawback of a limited field-of-view (FOV), as it can only measure objects directly in front of its image intensifier. As this intensifier is usually stationary, only movements that stay within the FOV can be measured. Thus, past research mainly examined the kinematics of the knee during tasks such as standing up from a chair, step-up/step-down, and deep knee bends [7]. When more extensive motions, such as gait, were investigated, only a portion of the motion could be captured. To overcome this limited FOV, our group at the Laboratory for Movement Biomechanics, ETH Zurich, developed a robotic single-plane fluoroscope that could autonomously follow the knee motion [4,7]. By (nearly) always keeping the knee in the FOV complete, consecutive cycles of level walking, downhill walking and stair descent could be measured with videofluoroscopy [13–20]. Our system only used the real-time estimates of the joint to achieve its joint position tracking, in contrast to other tracking methods that required one or more reference trials to be recorded before the tracking could be used in actual measurements [11]. This avoids the aforementioned reference trials, and tracking is possible immediately. Additionally, each trial is tracked independently from the others, which allows tracking of challenging tasks with high variability between trials, e.g. TKA patients performing level walking.

However, the automated single-plane fluoroscope could only track slow gait speeds [8,9], as it was limited by the traction between its wheels and the ground. The maximum imaging frequency of 30 Hz was too low to measure very fast movements, such as drop jumps, and a rigid support beam further restricted what activities could be measured. For example, stair ascent could not be investigated, as the beam would have driven into the stair. The tracking of the knee used a wire sensor connected to the subject's thigh, and while the effect of this wire was found to not measurably affect the subject's gait, subtle effects could not fully be excluded [9]. Lastly, the fact that images were acquired in a single plane reduced the achievable reconstruction of the out-of-plane knee kinematics—usually the kinematics in medio-lateral direction.

The desire for higher gait speeds, a wider choice of measurable movements, and improved kinematic reconstruction, has now led us to develop a next generation device: the tracking dual-plane fluoroscope (tDPF, Fig 1). By combining optical tracking with a bi-planar X-ray system on mechanically independent source and intensifier carriages on rails, the previous limitations of the wire sensor, single-plane acquisition, wheels, and rigid connection were addressed. The tDPF thus allows to directly measure the kinematics of the tibio-femoral joint *in vivo* during dynamic daily activities, including level walking and stair ascent/descent, at all gait speeds.

This work presents the tDPF and evaluates its capability of tracking the gait of healthy young adults during daily walking activities by assessing how well the knee centre stayed in the simulated intensifiers' FOVs when the subjects could move without speed limitations at self-selected speeds.

## Materials and methods

### The tracking dual-plane fluoroscope (tDPF)

The tDPF consists of two mechanically independently moving carriages equipped with a bi-planar X-ray system (Fig 1). An optical tracking system determines the position of the knee centre using an active diode fixed to the subject's thigh directly above the knee. A model-predictive control (MPC) controller moves the carriages so the X-ray system remains aligned with the knee centre. The closest knee to the image intensifiers is measured.

In the following, we will call the primary movement direction of the tDPF 'horizontal' and the perpendicular direction 'lateral' (the tDPF is not actuated in lateral direction). The direction perpendicular to 'horizontal' and 'lateral' is called 'vertical'.

**Mechanical system.** The carriages are horizontally driven by four synchronous motors with absolute position encoders (1FK7105-2AF71-1RH0, Siemens Switzerland AG). The motors transmit the propulsion through four planet-gearboxes (NRHP140-4-A2-z40-m3.183-TP-P1-AM, Güdel AG, Switzerland) onto a helical toothed rack-and-pinion drive. The horizontal movement can reach a maximum speed of $5\,\mathrm{m\,s^{-1}}$ and a maximum acceleration of $20\,\mathrm{m\,s^{-2}}$. The vertical axes are driven by two synchronous motors (1FK7101-2AC71-1RH1, Siemens Switzerland AG) that actuate two ballscrews (W4018G, NSK Deutschland GmbH, Germany) to move the fluoroscopic components vertically. The vertical movement can reach a maximum speed of $2\,\mathrm{m\,s^{-1}}$ and a maximum acceleration of $20\,\mathrm{m\,s^{-2}}$.

**X-ray system.** The X-ray system consists of two X-ray tubes (G1082, Varian Medical Systems, USA), two collimators (R 225 ACS, Ralco, Italy), and two X-ray image intensifiers (TH 9447 QX H694 L VR70, Thales Electron Devices S.A., France) connected to two high-speed

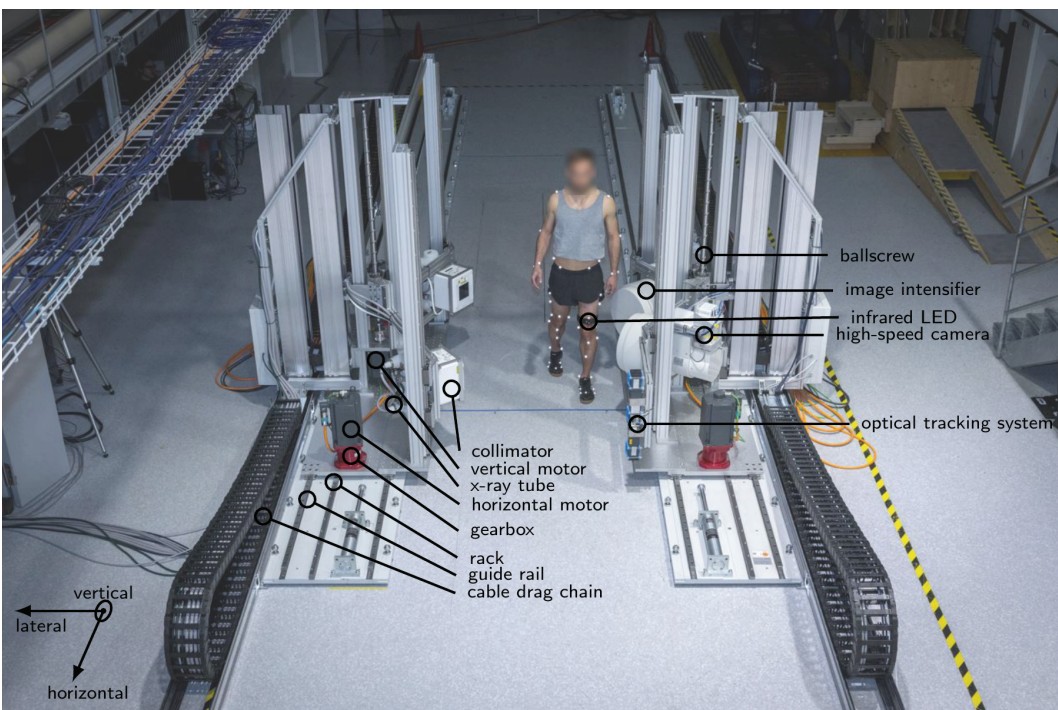

**Fig 1. Tracking dual-plane fluoroscope (tDPF).** The tDPF is a bi-planar X-ray system mounted on moving carriages, capable of tracking complete cycles of knee movement during level walking, ramp ascent/descent, and stair ascent/descent.

cameras (Phantom VEO E-340L, Vision Research, USA). The two X-ray planes are mounted at an angle of 70 degrees (Fig 2), with a source-intensifier distance of 1800 mm and a distance of 390 mm between the midpoint of the capture volume and the intensifier planes. As the radius of the intensifier sensor is 158 mm (experimentally measured), this setup (Fig 2) results in a capture volume with a length of 305 mm in horizontal direction and a length of 445 mm in lateral direction, and a length of 248 mm in vertical direction at the intersection of the beam cone symmetry axes (=capture volume centre).

**Optical knee tracking.** The knee center is tracked using an active marker tracking system (Fig 3): An infrared LED is mounted on an elastic strap that gets attached to the thigh just above the tracked knee at a comfortable and secure location on the thigh. Directionality of knee flexion/extension was not included in the tracking. To track something else than the LED, an offset can be provided to the system—for example, by measuring the offset between the knee centre and the LED using a folding rule the system will track the knee centre. The light from the infrared LED is captured by an infrared tracker (Visualeyez III VZ10K5 compact 3D motion tracker, PTI Phoenix Inc., Canada), and band-pass filters (BP735, Midwest Optical Systems, USA) prevent interference from the motion capture system (see below), which uses similar wavelengths. The sampling frequency of the infrared tracker is 6 kHz, and the data transfer frequency is limited to 500 Hz, i.e. 12 position vectors are obtained every 2 ms. These 12 position vectors are processed on a real-time processor (PXIe-1071, National Instruments Switzerland GmbH): First, outliers are detected whenever the difference between the measured horizontal position and the last horizontal position used by the controller

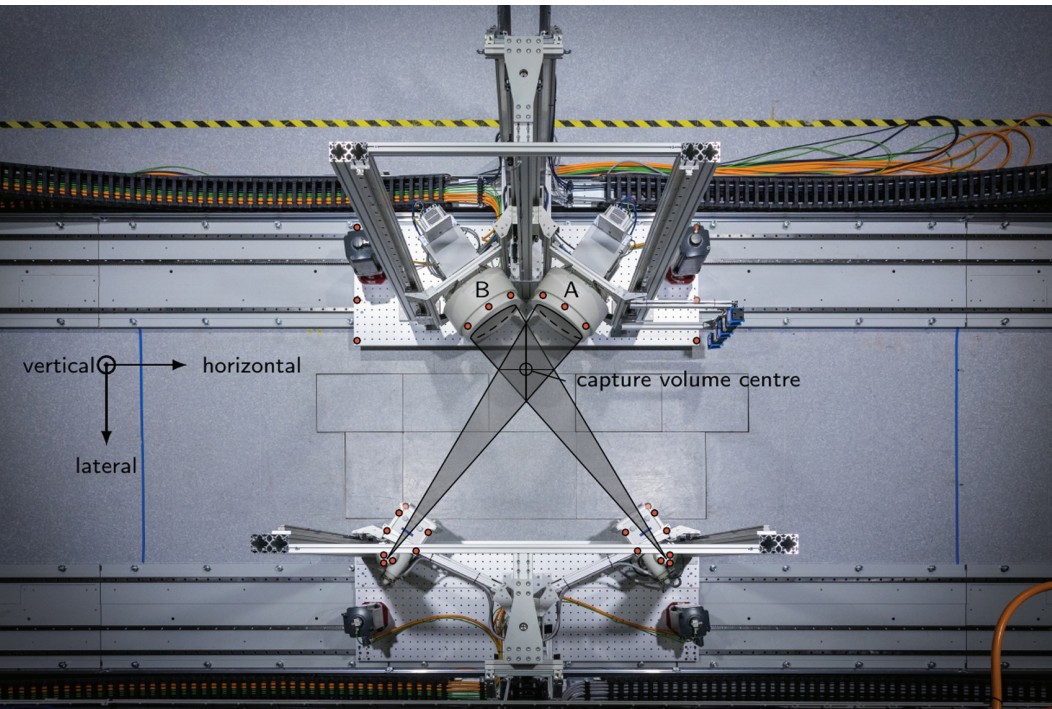

**Fig 2. X-ray system.** The capture volume has a length of 305 mm in horizontal direction and a length of 445 mm in lateral direction, along the cone symmetry axes, and a length of 248 mm in vertical direction at the intersection of the symmetry axes (=capture volume centre). The red points represent passive reflective markers used for measuring the position of the X-ray system.

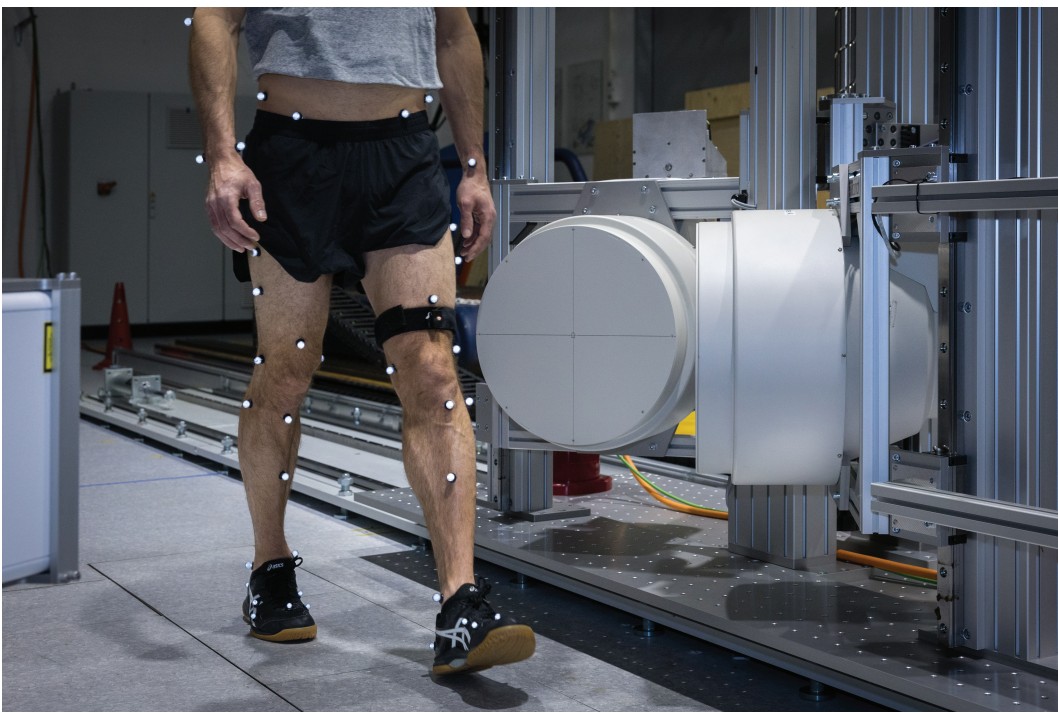

**Fig 3. Active marker to track the knee position.** The active marker mounted on an elastic strap (black band above the subject's left knee) emits infrared light, which is captured by a infrared tracker depicted in Fig 3. The passive reflective markers for the measurement of whole-body kinematics using optical motion capture.

exceeded 50 mm. Such a shift in position within 2 ms would imply a physically impossible position trajectory, and the corresponding positions are rejected. Second, the rejected positions are imputed by the following procedure: An estimate of the current position of the knee centre is calculated using a weighted least squares method over all non-outlier position values (weight of 1) and the previous position value (weight of 100). The previous position value is weighted stronger, as this value was processed in the previous loop, and thus it is chosen to be fixed. Thirdly, the estimate of the current position of the knee centre is smoothed using cubic splines (with smoothing parameter of 0.01) over the seven past positions of the knee centre.

This smoothed position is transmitted to the fail-safe software CPU (CPU1507S F, Siemens Switzerland AG), which controls the drive system. When the active marker signal is lost, an error handling routine stops the horizontal axis within a brake distance of less than 0.8 m and the vertical axis within less than 0.2 m if the marker signal remains lost. If the signal re-appears during the braking process and the distance between knee centre and capture volume centre is below 0.1 m, the braking procedure is cancelled and the axes recover towards the new reference position and speed values to re-align the capture volume centre with the knee centre.

**Tracking controller.** The two horizontal and two vertical axes are controlled independently by one model-predictive control (MPC) controller. Each controller aims to minimise the differences in position and velocity between capture volume centre and the (tracked) knee centre, i.e., the capture volume movement should track the knee centre. Based on the position error $e_{\mathrm{p}} = p_{\mathrm{k}} - p_{\mathrm{m}}$ between knee position $p_{\mathrm{k}}$ and motor position $p_{\mathrm{m}}$, and the velocity error

$e_v = \dot{p}_k - \dot{p}_m$, the controller finds an optimal motor torque $T_m^*$ that minimises these errors while ensuring that position, velocity, and motor torque remain within admissible ranges $\mathcal{P}$, $\mathcal{V}$, and $\mathcal{T}$:

$$
\begin{aligned}
e_p &= p_m - p_k \\
e_v &= \dot{p}_m - \dot{p}_k \\
T_m^* &= \text{argmin } f(T_m, e_p, e_v) \\
\text{s.t. } e_p &\in \mathcal{P} = [-100\,\text{mm}, 100\,\text{mm}] \\
e_v &\in \mathcal{V} = [-1\text{m/s}, 1\text{m/s}] \\
T_m &\in \mathcal{T} = [-290\,\text{N m}, 290\,\text{N m}]
\end{aligned}
\tag{1}
$$

Here, $\mathcal{T}, \mathcal{P}, \mathcal{V}$ are chosen so that the maximally allowed acceleration of $20\,\text{m s}^{-2}$ for individual components is not exceeded, and so that the capture volume centre closely tracks the knee. The controller feedback-loop is executed every 2 ms.

## Tracking capability study on 16 healthy subjects

To evaluate the tracking capability of the tDPF, an optical marker system compared the motion of tDPF with the motion of the knee by measuring reflective markers attached to the X-ray components and the subject's knee. By projecting the knee centre onto the image intensifiers, we evaluated how well the knee centre stayed in the simulated intensifiers' FOVs when the subjects could move without speed limitations at self-selected speeds.

16 subjects (five female and eleven male, average age of $29.8 \pm 5.7$ y and average BMI of $24.6 \pm 3.6\,\text{kg m}^{-2}$), without any prior experience using a tracking fluoroscope and without any acute or chronic musculoskeletal or neurological disorders participated in this study. Each subject performed five motion tasks (level walking, ramp ascent and descent, and stair ascent and descent) with the tDPF tracking their movement. Simultaneously, the motion of the subject and the motion of the tDPF were recorded by the motion capture system and the ground reaction forces were measured by force plates. The fluoroscopic systems of the tDPF were not used during this study and no ionizing radiation was created. The participants provided informed written consent, and this study was approved by the ETH Zurich Ethics Commission (EK-2023-N-69). The recruitment started on 14/05/2023 and ended on 09/06/2023.

**Motion tasks.** The five motion tasks were always performed in the same order: level walking, ramp ascent, ramp descent, stair descent, and stair ascent. During level walking, ramp ascent, and stair ascent, the left knee was tracked (as depicted in Fig 3). For ramp and stair descent the right knee was tracked, and thus the driving direction was reversed. The ramp measured $3100 \times 800 \times 540$ mm in length, width, and height, resulting in an inclination of $10°$. The stair consisted of three steps with dimensions $280 \times 800 \times 180$ mm in depth, width, and height (Fig 4).

During level walking, the vertical axis controller was switched off, i.e. the vertical axis remained at a constant height, which was aligned with the subject's knee height before the measurements. During the other activities vertical and horizontal controllers were active.

For each motion task the subject completed three trials to become familiarized with the situation and the measurement protocol. Then trials were measured until five valid trials were collected for each task, resulting in a total of 25 trials per subject. Following standard movement protocols, a trial was considered valid if at least one full valid gait cycle of the tracked knee was measured, if the first heel strike (HS) and following stance phase fell on one force plate (FP), and if the second HS fell on another FP. Here, one gait cycle is defined from HS

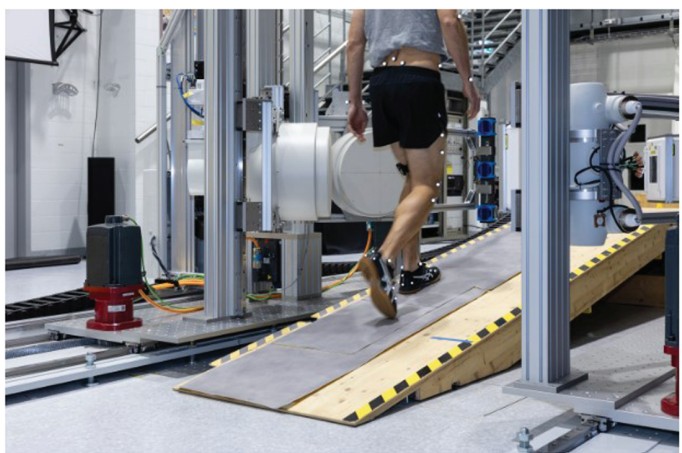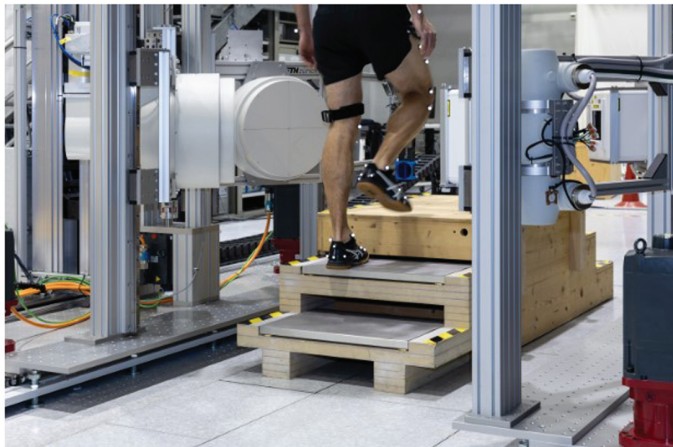

**Fig 4. Ramp and staircase instrumented with two mobile force plates.** The ramp measured $3100 \times 800 \times 540$ mm in length, width and height respectively, resulting in an inclination of $10°$. Each step of the stair measured $280 \times 800 \times 180$ mm in depth, width and height respectively.

to HS. Details on the different protocols used for the individual motion tasks can be found in Supporting information.

The measurements took about three to four hours for each subject, preparation not included. The subject had breaks of five to ten minutes between each motion task, as the measurement setup—including ramp, stair, and mobile FPs—had to be set up and/or calibrated.

**Motion capture and ground reaction force measurements.** 55 reflective skin markers were attached to the subject's skin at anatomical landmarks using adhesive tape (marker-set of the Institute for Biomechanics, ETH Zurich, see S1 Fig) and 28 reflective markers were mounted on various components of of the tDPF (Fig 2). 14 motion capture cameras (Vero, Vicon Motion Systems Ltd, UK) measured these markers' positions with a sampling frequency of 200 Hz. Simultaneously, ground reactions forces were assessed by force plates. For level walking, the five static force plates ($2 \times$ Type 9281B, $2 \times$ Type 9285, $1 \times$ Type 9281C; Kistler, Winterthur, Switzerland) were mounted on a separate concrete pillar between the rails of the tDPF system, and were thus mechanically decoupled from the surrounding ground and the tDPF. For stair or ramp tasks, the stair and ramp was screwed to the same decoupled concrete pillar and two mobile force plates (Type 9286AA, Kistler, Winterthur, Switzerland) measured the ground reaction forces. For ramp ascent and descent the two mobile plates were put on the ramp at a distance of 490 mm from the base, and for stair ascent and descent the two mobile force plates were placed directly onto the lower two steps of the stair.

The position of the subject's trunk $s_n$ was determined using the sacrum marker and the gait speed was calculated using a first-order backward difference $v(t_n) = (s(t_n) - s(t_{n-1})) \cdot f$, with $f = 200$ Hz denoting the sampling frequency. We then calculated the arithmetic mean and the standard deviation of the gait speed in horizontal and vertical directions over all frames for each subject and motion task.

The position of the knee centre was defined as the midpoint of the line connecting lateral and medial epicondyle markers. For context, we also determined the maximum speed and maximum acceleration of the knee centre for each subject and each task: speed and acceleration were calculated using a first-order backward difference, respectively. The resulting time series were low-pass filtered using a third-order Savitzky-Golay filter with a window length

of 17 data points for speed and 25 data points for acceleration. Mean and SD were calculated over all 16 subjects and valid gait cycles per motion task.

**Tracking performance metrics.** The positions of the components of the bi-planar X-ray system were determined using markers attached to the image intensifiers, collimators, and X-ray sources. Based on these positions, simulated radiation cones were calculated to mimic the radiation field of the X-ray system. The cones' symmetry axes were determined by calculating the midpoints of the image intensifiers and collimators through averaging the markers on the individual components. The radii of the simulated cones were set to 158 mm, which corresponds to the active area of the image intensifier. Using these simulated radiation cones, we projected the knee centre position onto the image intensifiers.

Similarly to the calculation for the knee centre above, we calculated the maximum speed and acceleration of the capture volume centre—using first-order backward differences and a third-order Savitzky-Golay filter with a window length of 17 data points for speed and 25 data points for acceleration—for each subject and trial and present the mean and SD over all subjects and valid gait cycles for each motion task.

Tracking performance was quantified using two metrics: First, the numbers of frames for which the knee centre remained in no, one, or both simulated FOV(s), summed up over all frames of all 16 subjects and valid gait cycles per motion task. Second, percentile plots of the distance between the knee centre projection (onto each image intensifier) and the intensifier midpoint visualised how closely the intensifiers were able to track the knee. Percentile plots are presented over all frames and valid gait cycles, per motion task and per subject.

## Results and discussion

This study used a motion capture system to evaluate how well the tDPF tracked the knee for different gait activities (level gait, stair ascent/descent, ramp ascent/descent) in young healthy subjects. While our old system was limited to a tracking speed of below $1 \, \text{m s}^{-1}$ [9], the tDPF allows subjects to choose their speed freely. The average gait speed in this study (mean and SD of $1.34 \pm 0.14 \, \text{m s}^{-1}$ during level walking, see Table 1) was consistent with values observed in literature for healthy subjects [10,12]. The lowest gait speed of a subject during a single trial in our study was $0.94 \, \text{m s}^{-1}$, which is similar to the typical gait speeds observed in TKA patients ($0.94 \pm 0.11 \, \text{m s}^{-1}$ measured in [8]). The range of gait speeds within this study thus covered both slow and fast walking speeds.

For level walking and ramp ascent, the tDPF was able to track the knee perfectly by keeping the knee centre in the simulated FOV of both intensifiers regardless of the subject's self-selected gait speed (see Table 2). For ramp descent, stair ascent, and stair descent, the tracking was still excellent, with the knee staying in the simulated FOV of both intensifiers for > 99%

**Table 1. Gait, knee centre, and capture volume centre metrics.**

| | Gait | | Knee centre | | | | Capture volume centre | | | |
|---|---|---|---|---|---|---|---|---|---|---|
| | $\|v\|$ (m/s) | | $\|v\|_{\text{max}}$ (m/s) | | $\|a\|_{\text{max}}$ (m/s$^2$) | | $\|v\|_{\text{max}}$ (m/s) | | $\|a\|_{\text{max}}$ (m/s$^2$) | |
| | hor | ver | hor | ver | hor | ver | hor | ver | hor | ver |
| Level walking | $1.34 \pm 0.14$ | | $2.79 \pm 0.21$ | | $18.73 \pm 5.31$ | | $3.06 \pm 0.21$ | | $17.45 \pm 3.71$ | |
| Ramp ascent | $1.17 \pm 0.12$ | $0.22 \pm 0.03$ | $2.59 \pm 0.26$ | $0.95 \pm 0.08$ | $12.78 \pm 1.93$ | $7.54 \pm 1.98$ | $2.84 \pm 0.26$ | $1.37 \pm 0.15$ | $12.81 \pm 2.24$ | $9.76 \pm 2.49$ |
| Ramp descent | $1.25 \pm 0.15$ | $0.22 \pm 0.03$ | $2.30 \pm 0.27$ | $0.66 \pm 0.09$ | $14.02 \pm 3.66$ | $8.44 \pm 2.13$ | $2.49 \pm 0.31$ | $0.87 \pm 0.15$ | $15.11 \pm 3.28$ | $12.68 \pm 2.78$ |
| Stair ascent | $0.67 \pm 0.07$ | $0.24 \pm 0.03$ | $1.85 \pm 0.17$ | $0.74 \pm 0.09$ | $12.60 \pm 2.84$ | $9.94 \pm 3.40$ | $2.12 \pm 0.20$ | $0.98 \pm 0.14$ | $11.68 \pm 2.52$ | $10.04 \pm 2.76$ |
| Stair descent | $0.88 \pm 0.12$ | $0.28 \pm 0.05$ | $1.99 \pm 0.38$ | $0.94 \pm 0.08$ | $15.90 \pm 4.98$ | $10.16 \pm 1.97$ | $2.03 \pm 0.34$ | $1.04 \pm 0.12$ | $11.30 \pm 2.63$ | $11.54 \pm 2.86$ |

Arithmetic mean and SD over all frames of all 16 subjects and valid gait cycles. During level walking, the vertical axis controller was disabled, i.e. the vertical axis remained at a constant height, which was aligned with the subject's knee height before the measurements.

**Table 2. Tracking capability metrics.**

|  | in no FOV | | in one FOV | | in both FOVs | |
|---|---|---|---|---|---|---|
|  | n | % | n | % | n | % |
| **Level walking** | 0 | 0 | 0 | 0 | 18223 | 100 |
| **Ramp ascent** | 0 | 0 | 0 | 0 | 17674 | 100 |
| **Ramp descent** | 0 | 0 | 5 | 0.03 | 16854 | 99.97 |
| **Stair ascent** | 0 | 0 | 9 | 0.05 | 19858 | 99.95 |
| **Stair descent** | 0 | 0 | 84 | 0.52 | 15961 | 99.48 |

FOV radius = 158 mm. The numbers of frames in which the knee centre remained in no, one, or both simulated FOV(s) during a valid gait cycle, summed up over all frames of all 16 subjects and valid gait cycles per motion task.

of frames. Importantly, the remaining frames could still be registered through single-plane registration as the knee centre always stayed in at least one simulated FOV. These results confirm that the tDPF allows tracking of the knee for level walking and ascent/descent tasks over the full range of gait speeds, and provide the basis for successful 3D reconstruction of knee kinematics using bi-planar fluoroscopy throughout complete gait cycles.

While the tracking performance during this experiment was excellent, Fig 5 shows that tracking varied substantially between subjects within a task. As the distribution of velocities and accelerations were very similar between subjects within a task (see S2 and S3 Figs), these are unlikely to be the source for these differences in tracking. Therefore there needs to be another effect causing differences in performance tracking between subjects.

The polar percentile plots of the knee centre projected onto the intensifier (Fig 6) show that the tracking pattern is task-dependent and asymmetric. Interestingly, the pattern of intensifier A is shifted to the left for nearly all tasks while that of intensifier B is shifted to the right. For level walking, stair ascent, and ramp ascent, one could expect both patterns to mostly lie in the left part of the intensifier, because the subjects were moving from left to right during these tasks with the tDPF trailing this movement. For stair descent and ramp descent, the subjects were walking from right to left, so one would expect all patterns to be shifted to the right.

However, this effect seems to be overshadowed by another geometric effect, which arises from the lateral distance between the knee centre and both image intensifiers. If a subject walks farther away from the image intensifiers, the projection of the knee centre will move to the outside of both intensifiers (Fig 7): left for intensifier A and right for intensifier B—which is what we observe. For further confirmation, we calculated the Pearson correlation coefficients between the average horizontal knee position when projected onto the intensifier (Fig 6) and the lateral distance between knee centre and the image intensifiers (Fig 8). These correlations ranged between 0.60 and 0.87 for intensifier A depending on task, and between −0.46 and −0.83 for intensifier B (see the Supporting information), indicating that the distance between subject and the intensifiers indeed affects the tracking and can explain the observed asymmetry in tracking performance.

As the lateral distance between knee centre and image intensifiers only affects the horizontal tracking, the asymmetries in the vertical direction must have had a different cause. Indeed, inspection of the vertical knee-centre position (Fig 8) shows differences of up to 150 mm between subjects, which is primarily caused by a safety restriction. As described above, the subject-specific knee-height was determined at the beginning of the experiment and used as the vertical set-point for the tracking during all tasks. However, the knee height of the shortest subjects was lower than the minimum safety distance to the floor. Whenever the knee centre was lower than this safety distance, the tDPF would stop tracking and keep at the minimum vertical height, leading to the negative values in the vertical plots of Fig 8. Particularly

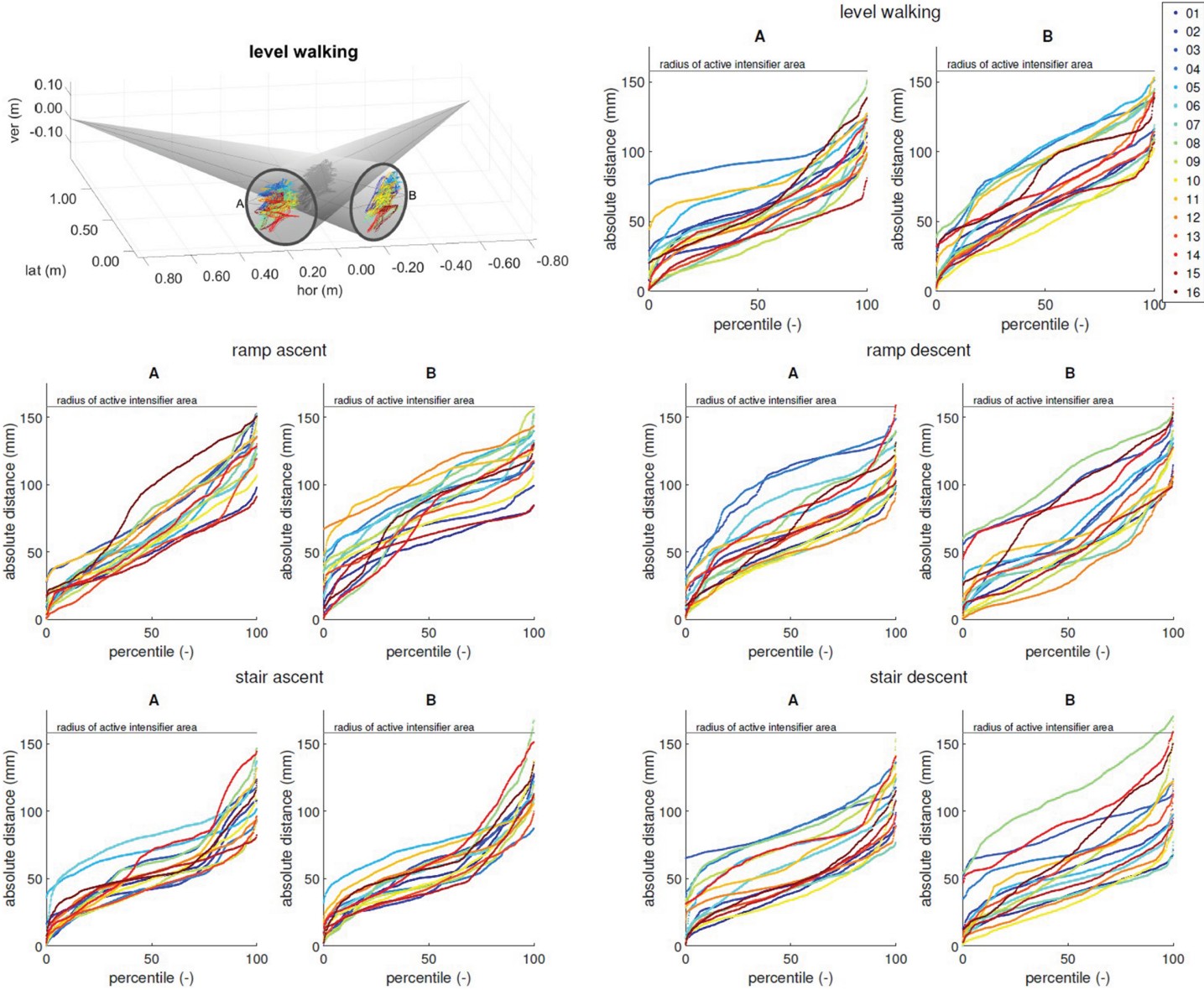

**Fig 5. Absolute distance of the knee centre's projection.** The absolute distance of the knee centre's projection on simulated FOV A and B to the FOV centres (top left shows the projections for level walking) are shown in percentiles, over all frames and valid gait cycles, per motion task and per subject. For level walking, stair ascent, and ramp ascent, the subjects were moving from left to right, while for stair descent and ramp descent, the subjects were walking from right to left.

for short subjects during level gait, this was a common occurrence. This safety restriction was necessary as the experiments were performed in a prototype laboratory where rails where bolted onto the floor, and will be unnecessary in our research laboratory, where the rails are countersunk with the floor.

The maximum accelerations performed by the tDPF (see Table 1) while tracking level gait were close to the system's maximum acceleration of $20\,\mathrm{m\,s^{-2}}$, which is limited by the technical specifications of the cable drag chain. This means that for movements with much higher accelerations the tracking performance will diminish and the knee centre might leave

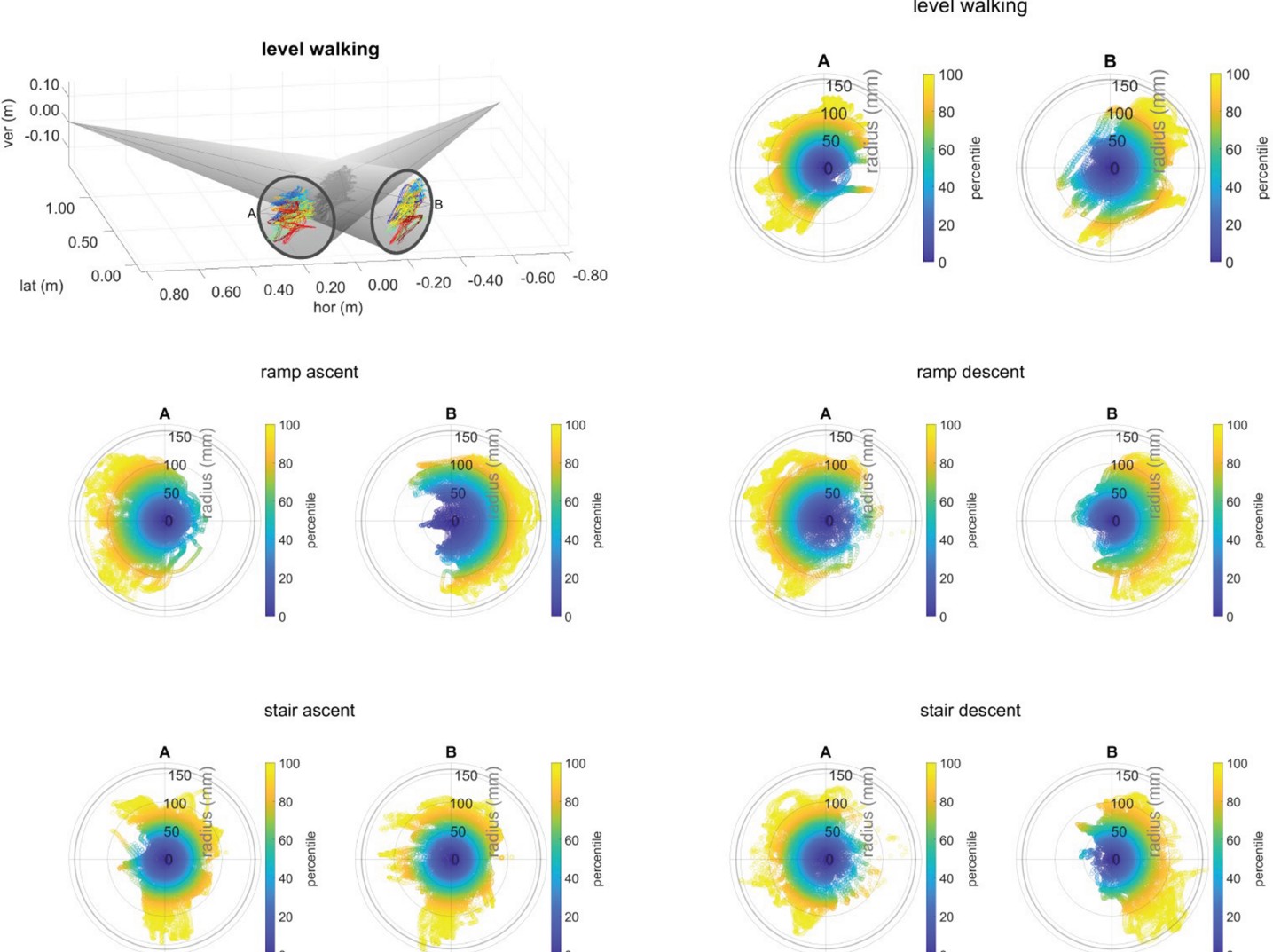

**Fig 6. Percentile plots of the knee centre's projection** The position of the knee centre projected onto the image intensifiers are shown for each motion task in percentiles over all frames, valid gait cycles, and subjects. For level walking, stair ascent, and ramp ascent, the subjects were moving from left to right, while for stair descent and ramp descent, the subjects were walking from right to left. Note the asymmetric shift to the left for intensifier A and to the right for intensifier B.

both FOVs for some frames. As the tracking still seems to have some reserves (about 80% of frames are within 10 cm of the image intensifier's midpoint, see Fig 6), tracking of movements with slightly higher accelerations (e.g. jogging) probably remains possible, especially if subjects can be made to walk with the optimal lateral distance between knee and intensifiers. While we consider adding a lateral tracking infeasible due to the mechanical challenges and safety risks, other guiding methods, such as adding a line on the floor or providing feedback to the subject after each trial, might be worthwhile. However, such measures might impact the subject's gait, and would need to be investigated separately. Ramp ascent, ramp descent, stair ascent, and stair descent required lower accelerations than level gait, and thus tracking faster executions of these tasks should pose no problems. Additionally to tracking slow and fast knee movements in TKA patients, the use of the tDPF could be extended to other

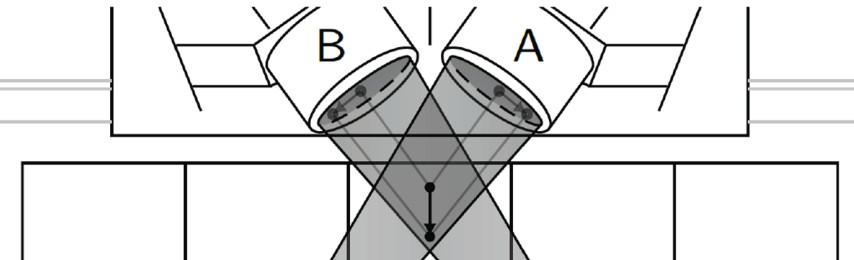

**Fig 7. Shift in projection.** When the lateral distance between knee centre and image intensifier increases, the projections on the image intensifiers shift to the outside of the intensifiers.

movement disorders such as cerebral palsy (CP). By tracking the knee during crouch gait, insight can be won into CP subjects' tibio-femoral kinematics which could support treatment planning. The application area of the tDPF could also be extended to other joints of the human body—e.g. patello-femoral joint, hip, or shoulder—due to the range in vertical travelling distance. Thereby enabling not only research towards implant development and healthy kinematics but also pathogenesis of for example shoulder problems in wheelchair users [22].

This study is limited by the fact that the X-ray system was not used due to ethical considerations, as the purpose of this study was to confirm tracking performance prior to exposing participants to X-rays. Thus, it remains to confirm that the resulting image quality—possibly affected by non-complete captures of the knee and vibrations on the structures carrying the X-ray system—would be sufficient to perform kinematic reconstruction of the knee with the required accuracy. A previous publication using a single-plane fluoroscope indicated that a partial image of the knee including its centre was sufficient to reconstruct the knee kinematics [9]. While the driving behaviour of tDPF remained stable during the study, the high accelerations could have possibly caused small deformations on the structure carrying the X-ray units that were too small to detect by the optical motion capture system. The impact of such deformations on the resulting X-ray images needs to be investigated. It is further conceivable that the presence and operation of the tDPF causes the subjects to move differently. This effect could not be detected for the previous moving single-plane fluoroscope [9], despite the wire sensor attached to the subject's knee and the C-arm connecting the two carriages in front of the subject potentially posing a visual distraction. While the new system uses optical tracking and does not feature any mechanical structures directly in front of the subject, the device itself is taller and might thus be more imposing. The impact of the tDPF on gait characteristics will be analysed in a next study.

## Conclusion

This study presented a novel tracking dual-plane fluoroscope to track the knee joint of young, healthy subjects during level walking, ramp ascent, ramp descent, stair ascent, and stair descent at self-selected gait speeds. The ability to keep the knee centre in the simulated FOV of both image intensifiers has been demonstrated in 16 healthy subjects, using skin marker measurements of the knee joint position. To the authors' knowledge, the tDPF presented in this paper is the first dual-plane fluoroscope that was able to track the knee joint during entire cycles of stair and ramp ascent at self-selected gait speeds for young and healthy subjects. Further, our device does not require any pre-recording of movement patterns and only uses

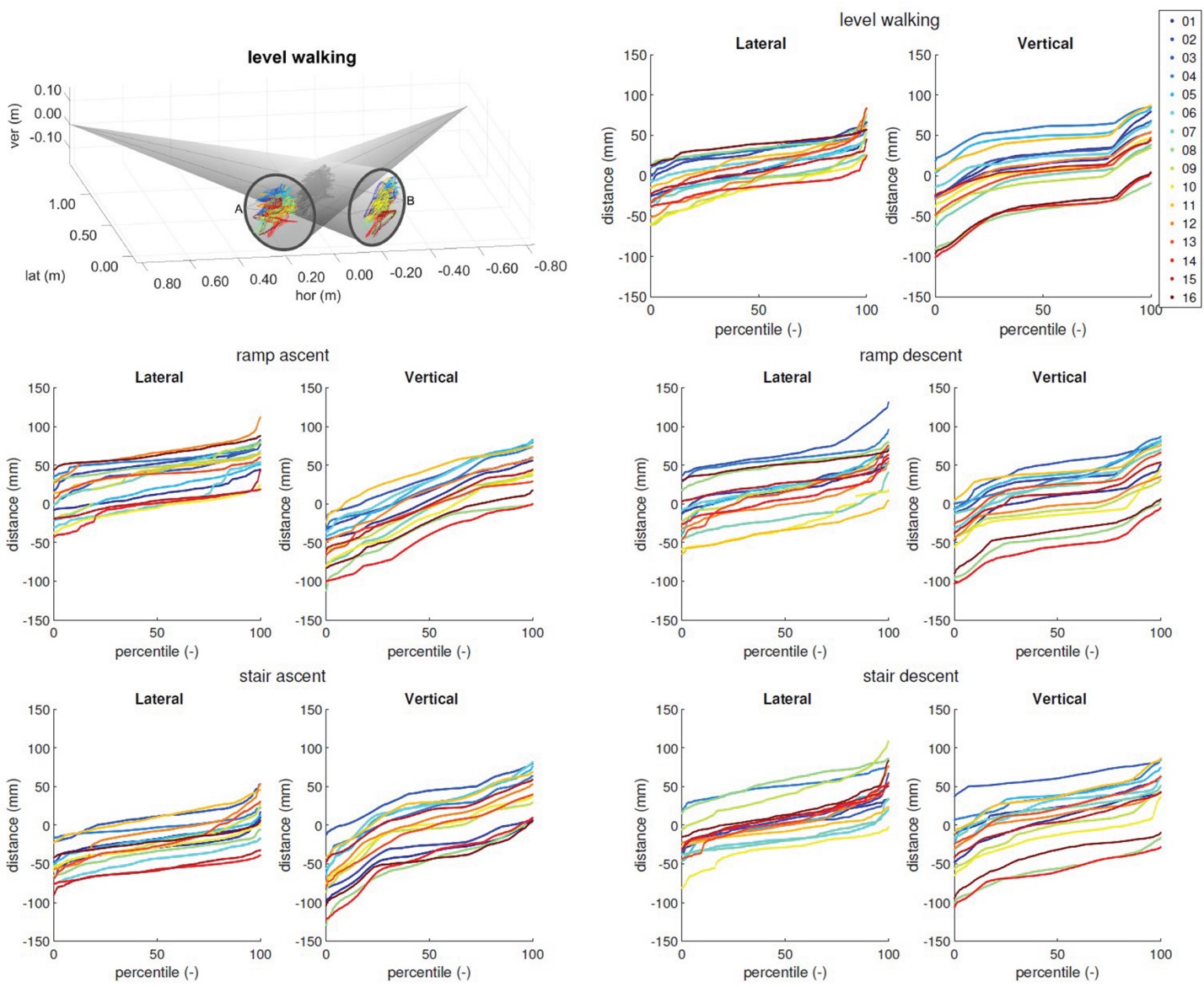

**Fig 8. Distance between knee centre and capture volume centre.** For level walking, stair ascent, and ramp ascent, the subjects were moving from left to right, while for stair descent and ramp descent, the subjects were walking from right to left. An increasing value in lateral direction indicates an increasing distance between knee centre and image intensifiers. An positive value in vertical direction indicates that the knee centre is above the capture volume centre.

real-time position estimates of the tracked joint. By operating independently on each trial—without relying on prior trial data—our system is inherently suited to accommodate inter-trial variability, which is especially relevant in clinical contexts where such variability is common. While not directly assessed in this study, this robustness is a core feature of the tracking capability and supports broader applicability in patient populations.

## Supporting information

**S1 Fig. Marker-set of the Institute for Biomechanics, ETH Zurich.**
(PDF)

**S2 Fig. Speed of knee centre.** For level walking, stair ascent, and ramp ascent, the subjects were moving from left to right, while for stair descent and ramp descent, the subjects were walking from right to left.
(PNG)

**S3 Fig. Acceleration of knee centre.** For level walking, stair ascent, and ramp ascent, the subjects were moving from left to right, while for stair descent and ramp descent, the subjects were walking from right to left.
(PNG)

**S1 Table. Pearson correlation coefficients.**
(JPG)

**S1 Description.**

**Level walking.** By observing the familiarization trial, the study leader adjusted the subject's starting distance from the FPs to ensure that a HS of the left foot would fall cleanly onto FP2 during the subject's natural gait. Subjects were told to start with the left foot and ignore the FPs while walking. The measurement team checked the validity of the gait cycles during every trial.

**Ramp ascent.** The subject started to walk with their left foot, left HS on upper FP, first step was—dependent on the step length—fully or only partially on the ramp.

**Ramp descent.** The subject started with their right foot, right HS on upper FP, start was directly at the beginning of the ramps slope, the FPs were reached on the third step.

**Stair descent.** The subject started with their right foot, right HS on lower FP, start was on the top step of the stair, first step with right foot on the top step.

**Stair ascent.** The subject started with their right foot, left HS on upper FP, two starting steps before climbing the stair (right left).

## Acknowledgments

We would like to extend our heartfelt gratitude to Peter Schwilch, Charlotte Lang, Martin Bertsch, and Demian Siegwart for their invaluable contributions and support throughout this study.

## Author contributions

**Conceptualization:** Albert Planta, Pascal Schütz.

**Data curation:** Albert Planta, Raphael Surbeck.

**Formal analysis:** Albert Planta.

**Investigation:** Albert Planta, Raphael Surbeck, Pascal Schütz.

**Methodology:** Albert Planta, Raphael Surbeck, Stefan Plüss, Pascal Schütz.

**Project administration:** Albert Planta, Raphael Surbeck.

**Resources:** Albert Planta, Stefan Plüss, Pascal Schütz.

**Software:** Albert Planta.

**Supervision:** William R. Taylor, Pascal Schütz.

**Validation:** Albert Planta.

**Writing – original draft:** Albert Planta, Pascal Schütz.

**Writing – review & editing:** Albert Planta, Raphael Surbeck, Florian Vogl.

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
