## [Decision Letter · Decision Letter 0]

10 Jul 2025

PONE-D-24-27653A dual-plane fluoroscope to track joint kinematics during dynamic daily activitiesPLOS ONE

Dear Dr. Planta,

Thank you for submitting your manuscript to PLOS ONE. After careful consideration, we feel that it has merit but does not fully meet PLOS ONE’s publication criteria as it currently stands. Therefore, we invite you to submit a revised version of the manuscript that addresses the points raised during the review process.

I have struggled to secure a 2nd review for this article, and so have relied on the extensive review of the first reviewer. The reviewer was generally positive about the article but highlight areas that require more clarity and challenged some of the methods used. Please see comments below. 

We look forward to receiving your revised manuscript.

Kind regards,

Aliah Faisal Shaheen

Academic Editor

PLOS ONE

Journal requirements: When submitting your revision, we need you to address these additional requirements. 1. Please ensure that your manuscript meets PLOS ONE's style requirements, including those for file naming. The PLOS ONE style templates can be found at https://journals.plos.org/plosone/s/file?id=wjVg/PLOSOne_formatting_sample_main_body.pdf and https://journals.plos.org/plosone/s/file?id=ba62/PLOSOne_formatting_sample_title_authors_affiliations.pdf. 2. We note that the grant information you provided in the ‘Funding Information’ and ‘Financial Disclosure’ sections do not match.  When you resubmit, please ensure that you provide the correct grant numbers for the awards you received for your study in the ‘Funding Information’ section. 3. Please provide a complete Data Availability Statement in the submission form, ensuring you include all necessary access information or a reason for why you are unable to make your data freely accessible. If your research concerns only data provided within your submission, please write "All data are in the manuscript and/or supporting information files" as your Data Availability Statement.

Reviewers' comments:

Reviewer's Responses to Questions

**Comments to the Author**

1. Is the manuscript technically sound, and do the data support the conclusions?

Reviewer #1: Yes

2. Has the statistical analysis been performed appropriately and rigorously? 

Reviewer #1: Yes

3. Have the authors made all data underlying the findings in their manuscript fully available?

Reviewer #1: Yes

4. Is the manuscript presented in an intelligible fashion and written in standard English?

Reviewer #1: Yes

5. Review Comments to the Author

Reviewer #1: This paper introduces the novel iteration of a double fluoroscope mounted on a robot to measure accurate knee kinematics during dynamic activities. The study goal is to present the device and validate the knee position tracking during various activities with healthy participants. Such system is very relevant for the field and is very promising in terms of future applications. This is a high quality paper, well written and easy to follow. However, I still have two major concerns, the first one regarding the metrics used to assess the quality of the tracking and the second one on the shared data.

Major concerns:

Metrics - The metric used is the distance between the knee centre and the centre of the field of view (FoV) of each fluoroscope. The study shows multiple timepoints where the knee centre is on the edge of one or the other fluoroscope’s FoV. If the knee centre is at the edge of the FoV, the image measured by the fluoroscope will not contain the whole view of the knee, e.g. a femoral condyle or part of the tibial plateau may be missing. And, if this is the case on both views at the time the view of the anatomical structure may be quite limited.

My concern is the following: is it possible to estimate accurately the pose of the femur or tibia with only partial view of the bone on one or two images? If this is the case, the authors should elaborate on this in the manuscript with appropriate references. If this is not the case, the results may overestimate the quality of the tracking and authors may have to use a metric estimating whether the volume of the knee is within the volume of acquisition during the various tasks, instead of 2 separate 2D metrics.

Data - The authors made a great effort in publishing the data, however, they are currently in proprietary format and requires MATLAB licences to be used. This goes against the FAIR (Findability, Accessibility, Interoperability, and Reusability) Data Principles, please update the data format to match these recommendations.

Minor concerns & Comments:

General questions:

- Although this may not be the focus of research, could the authors elaborate on potential applications of this system to other joints?

- The study mention the tibio-femoral joint, could this tool be used to track the patello-femoral joint as well? If so, it would be interesting to mention it for reader not familiar with dual fluoroscopy.

- What is the weight of the full system? Did you have to reinforce the screed of the room where the system is?

- Since the emitters is on one side of the walkway, is possible to measure either knee when walking or can the system only measure the knee that is closest/farthest to the emitter?

- The title and legend of the figure seems to be missing. In PlosOne they are generally reported after the paragraph where they are mentioned first.

Specific remarks:

- P2L26: “our system only used the real-time estimates of the joint to achieve its tracking”. Do you mean “joint centre”? if so, please specify to help the understanding.

- P2L30: “tasks with high variability between trials”, could you provide 1 or more examples to clarify what you mean?

- P2L31: The “However” seems irrelevant, please rephrase.

- P3L99: How was the threshold of 50mm chosen?

- P4EQ1: The admissible range of the motor torque mentioned L123 is not reported.

- P5L165: The four hours include also preparation of setup and cleaning or just the time with the participant?

- P7L242: ”Therefore there need to be another effect causing differences in performance tracking between subjects”. Does the authors have some hypotheses on the other potential sources of errors?

- Figure 5 represent the distribution in percentage of the distance to the centre of the FoV?

- Are figure 10,11,12,13 mentioned in the manuscript?

6. PLOS authors have the option to publish the peer review history of their article (what does this mean?). If published, this will include your full peer review and any attached files.

Reviewer #1: **Yes: **Xavier Gasparutto

---

## [Author Response · Author response to Decision Letter 1]

20 Feb 2025

Responses to Reviewers

Reviewer #1: This paper introduces the novel iteration of a double fluoroscope mounted on a robot to measure accurate knee kinematics during dynamic activities. The study goal is to present the device and validate the knee position tracking during various activities with healthy participants. Such system is very relevant for the field and is very promising in terms of future applications. This is a high quality paper, well written and easy to follow. However, I still have two major concerns, the first one regarding the metrics used to assess the quality of the tracking and the second one on the shared data.

We thank the reviewer for his valuable input and constructive feedback. Answers are below in a point-to-point manner in red. Changes made to the manuscript are highlighted in yellow, the page and line numbers mentioned refer to the adapted manuscript with tracked changes (Manuscript_adapted_TC).

Major concerns:

• Metrics - The metric used is the distance between the knee centre and the centre of the field of view (FoV) of each fluoroscope. The study shows multiple timepoints where the knee centre is on the edge of one or the other fluoroscope’s FoV. If the knee centre is at the edge of the FoV, the image measured by the fluoroscope will not contain the whole view of the knee, e.g. a femoral condyle or part of the tibial plateau may be missing. And, if this is the case on both views at the time the view of the anatomical structure may be quite limited.

My concern is the following: is it possible to estimate accurately the pose of the femur or tibia with only partial view of the bone on one or two images? If this is the case, the authors should elaborate on this in the manuscript with appropriate references. If this is not the case, the results may overestimate the quality of the tracking and authors may have to use a metric estimating whether the volume of the knee is within the volume of acquisition during the various tasks, instead of 2 separate 2D metrics.

The reviewer raises an important issue that we have discussed intensively within our team. Based on the input from our colleagues who are working on the image-processing, it was decided that partial view of the knee (at least half of the knee is in the field of view) should be sufficient for the intended analysis of the data. Further, a similar metric was used in a previous publication using a single-plane fluoroscope (List et al 2017), in which they reconstructed the knee kinematics using the partial view of the knee including its centre A next study including imaging is however necessary to ensure this also holds in practise for our dual-plane fluoroscope. To better inform the reader about the possible limitations of the chosen metric, we added the following sentences within the discussion:

Page 9, Lines 308-313:

Thus, it remains to confirm that the resulting image quality – possibly affected by non-complete captures of the knee and vibrations on the structures carrying the X-ray system – would be sufficient to perform kinematic reconstruction of the knee with the required accuracy. A previous publication using a single-plane fluoroscope indicated that a partial image of the knee including its centre was sufficient to reconstruct the knee kinematics [9]

• Data - The authors made a great effort in publishing the data, however, they are currently in proprietary format and requires MATLAB licences to be used. This goes against the FAIR (Findability, Accessibility, Interoperability, and Reusability) Data Principles, please update the data format to match these recommendations.

Our .mat files can also be opened and processed within Python, please see for example https://www.askpython.com/python/examples/mat-files-in-python. Due to the nested structure of the .mat files we consider this clearer for the interested reader compared to providing .csv or .tx

Minor concerns & Comments:

General questions:

1. Although this may not be the focus of research, could the authors elaborate on potential applications of this system to other joints?

A paragraph on possible elaboration of the potential application of the system to other joints have been added in the discussion.

Page 9, lines 298-306:

Additionally to tracking slow and fast knee movements in TKA patients, the use of the tDPF could be extended to other movement disorders such as cerebral palsy (CP). By tracking the knee during crouch gait, insight can be won into CP subjects’ tibio-femoral kinematics which could support treatment planning. The application area of the tDPF could also be extended to other joints of the human body – e.g. patello-femoral joint, hip, or shoulder – due to the range in vertical travelling distance. Thereby enabling not only research towards implant development and healthy kinematics but also pathogenesis of for example shoulder problems in wheelchair users.

2. The study mentions the tibio-femoral joint, could this tool be used to track the patello-femoral joint as well? If so, it would be interesting to mention it for reader not familiar with dual fluoroscopy.

Within our current manuscript we focused on the application on the tibio-femoral joint. Of course the system could also be used to track the patella-femoral joint, this might however require other tracking optimizations. We added the possible application to the patella-femoral joint in the discussion.

See above.

3. What is the weight of the full system? Did you have to reinforce the screed of the room where the system is?

Approximately 642kg (source carriage) and 663kg (intensifier carriage). The system is mounted on concrete.

4. Since the emitters is on one side of the walkway, is possible to measure either knee when walking or can the system only measure the knee that is closest/farthest to the emitter?

Per walk, the closest knee to the image intensifiers can be measured. We added this information within the methods section.

Page 3, lines 59-60:

The closest knee to the image intensifiers is measured.

5. The title and legend of the figure seems to be missing. In PlosOne they are generally reported after the paragraph where they are mentioned first.

The title and legends of the figures are now added within the manuscript.

Specific remarks:

6. P2L26: “our system only used the real-time estimates of the joint to achieve its tracking”. Do you mean “joint centre”? if so, please specify to help the understanding.

Here, we specifically mean the joint and not the joint centre, as we describe the single-plane fluoroscope’s tracking, not the dual-plane fluoroscope’s tracking nor the metric used in this manuscript.

7. P2L30: “tasks with high variability between trials”, could you provide 1 or more examples to clarify what you mean?

An example has been added.

Page 2, line 30:

… , e.g. TKA patients performing level walking.

8. P2L31: The “However” seems irrelevant, please rephrase.

We think the “however” is relevant, as in the previous paragraphs the advantages of the single-plane fluoroscope are explained, while in this paragraph the limitations are elaborated. We however rephrased one of the above sentences to make that clearer.

Page 2, lines 24ff:

Our system only used the real-time estimates of the joint to achieve its tracking, in contrast to other tracking methods that required one or more reference trials to be recorded before the tracking could be used in actual measurements. This avoids the aforementioned reference trials, and tracking is possible immediately. Additionally, each trial is tracked independently from the others, which allows tracking of challenging tasks with high variability between trials, e.g. TKA patients performing level walking.

However, the automated single-plane fluoroscope could only track slow gait speeds, as it was limited by the traction between its wheels and the ground.

9. P3L99: How was the threshold of 50mm chosen?

This threshold was chosen heuristically.

10. P4EQ1: The admissible range of the motor torque mentioned L123 is not reported.

We added the information in the formula.

11. P5L165: The four hours include also preparation of setup and cleaning or just the time with the participant?

The four hours included just the time with the participant, this information has been added to the method section.

Page 5, lines 169-170:

…, preparation not included.

12. P7L242: ”Therefore there need to be another effect causing differences in performance tracking between subjects”. Does the authors have some hypotheses on the other potential sources of errors?

The other potential source of error is elaborated in the following paragraphs.

13. Figure 5 represent the distribution in percentage of the distance to the centre of the FoV?

Figure 5 shows the distribution (in percentiles) of the absolute distance between the knee centre and FoV centre per task and subject.

14. Are figure 10,11,12,13 mentioned in the manuscript?

All figures are now referenced within the manuscript. We removed the previous Figs 9 and 12. Figs 10, 11 and 13 are now called S1, S2 and S3 Fig.

---

## [Decision Letter · Decision Letter 1]

14 May 2025

PONE-D-24-27653R1A dual-plane fluoroscope to track joint kinematics during dynamic daily activitiesPLOS ONE

Dear Dr. Planta,

Thank you for submitting your manuscript to PLOS ONE. After careful consideration, we feel that it has merit but does not fully meet PLOS ONE’s publication criteria as it currently stands. Therefore, we invite you to submit a revised version of the manuscript that addresses the points raised during the review process. I appreciate that you have already gone over one round of reviews already, we had to ask a different reviewer to look at the manuscript as the original reviewer wasn't available and they have provided a constructive review with further suggestions for improvements (see below).  

We look forward to receiving your revised manuscript.

Kind regards,

Aliah Faisal Shaheen

Academic Editor

PLOS ONE

Reviewers' comments:

Reviewer's Responses to Questions

**Comments to the Author**

1. If the authors have adequately addressed your comments raised in a previous round of review and you feel that this manuscript is now acceptable for publication, you may indicate that here to bypass the “Comments to the Author” section, enter your conflict of interest statement in the “Confidential to Editor” section, and submit your "Accept" recommendation.

Reviewer #2: (No Response)

2. Is the manuscript technically sound, and do the data support the conclusions?

Reviewer #2: Partly

3. Has the statistical analysis been performed appropriately and rigorously? 

Reviewer #2: Yes

4. Have the authors made all data underlying the findings in their manuscript fully available?

Reviewer #2: Yes

5. Is the manuscript presented in an intelligible fashion and written in standard English?

Reviewer #2: Yes

6. Review Comments to the Author

Reviewer #2: The authors present a first-of-its-kind dual fluoroscope setup for tracking the entire gait cycle during walking and stair ascent/descent. The experimental advance is noteworthy, and the authors are commended for developing an impressive setup for tracking knee joint motion in healthy participants and later in patients following total knee arthroplasty (TKA). The manuscript provides sufficient detail to describe the system setup and, if others had the resources, replicate the work. (Of particular note is the description for the tracking of knee joint position and controller.)

On the other hand, the potential advance of this setup is clouded by the lack of gold standard measurements to compare tracking results against. Knee position tracking is an understandable target, but the authors’ setup may perform worse than single plane system or static dual fluoroscopy systems due to movement of the emitters and intensifiers.

Furthermore, the knee centre location was not observed on fluoroscope images, further shadowing the potential advancement described in the manuscript. For example, knee joint positions on the trailing edge of the fluoroscopes will have very little bone in the image for registration, but knee joint positions towards the leading edge will have much of the bone available for image registration.

Another concern is the simulated cones for determining the fluoroscopes’ overlap. While the authors are commended for performing this calculation, the edges of the image intensifiers do not always yield usable image data for image registration. An object with the simulated cones dimensions, or close to it, imaged with both fluoroscopes, even statically, would lend confidence to the authors’ calculations. In the manuscript, all references to staying in the fluoroscopes’ FOV needs to include that the FOV was simulated.

Also of concern, but less so, the references to ‘all gait speeds’ refers to a narrow, self-selected band of speeds in healthy participants. Patients following TKA may walk at slower speeds and, more importantly, with higher variability, which, again, brings into question the advancement described by the authors. While the authors acknowledged the lowest gait speed for a single trial, separating the trials into different groups would have made their point about variable speeds much stronger.

Please change all references to ‘tracking’ with ‘knee joint position tracking’, because in the field of fluoroscopy tracking often refers to model-based tracking. And accuracy of tracking typically refers to comparisons to a gold standard for joint translations and rotations.

Specific Comments

Abstract

As referenced in the general comments, please clarify that the FOV was simulated, and that no images were acquired.

Introduction

No comments. Well written.

Methods

For lines 82-85, “As the radius of the intensifier sensor is 158mm, this setup (Fig 2) results in a capture volume with a length of 305mm in horizontal direction and a length of 445mm in lateral direction, and a length of 248mm in vertical direction at the intersection of the beam cone symmetry axes …”, were these theoretical values? Or experimental measures? Oftentimes the image intensifiers do not have usable image data all the way to the edge. Either way, the authors are commended for performing and reporting these calculations/measurements.

On line 87, “The knee centre is tracked using an active marker tracking system (Fig 3):….” Was the active marker placed at a standard distance up the thigh? And did the tracking include an directionality to determine if the knee was bent or straight?

The ‘Optical knee tracking’ section is noteworthy for the details provided on the experimental implementation. The text describes potential ‘rejected’ positions. Do the authors have an estimate for how often this occurred?

Three to four hours for acquisition seems rather long if no fluoroscope images were acquired. As alluded to the in the last comment, did the authors experience setbacks with the system that would not be expected in a traditional motion capture setup?

Results & Discussion

Could the authors describe the results in Figure 5 in more detail? The text merely states that the tracking varied substantially between subjects. Also of note in Figure 5, it’s not clear what the y axis is plotting from the caption. Distance from the projected knee joint position to the center of the image intensifier?

Lines 278-279, “leading to the negative values in the vertical plots of Fig 5.” The values in the plots look all positive. Are the authors referring to the positions plotted on the image intensifiers? Also, why did this not lead to the knee joint center position being outside the FOV, if the knee centre was lower the safety distance?

Could the authors explain this statement in greater detail? Lines 307-308, “This study is limited by the fact that the X-ray system was not used due to ethical considerations.” Isn’t the point of the system to use x-rays? It would be understandable to assess tracking of the knee joint position to confirm performance prior to exposing participants to x-rays. But it’s not clear if that was what the authors are implying.

Figures and Tables

Figure 1 is a nice zoomed-out view of the system, but, being the first figure, labels for the different components would benefit the readers who are not as familiar with fluoroscope systems.

The authors are applauded for the impressive polar plots in Figure 6 that clearly demonstrated the results.

For Figures 5, 6, 8, 10, and 11, it would help the reader interpret the images to know which direction the participants were moving in. This was mentioned in the text but not in the captions nor figures.

Conclusions

Same as the abstract, please insert ‘simulated’ prior to FOV, and clarify that no x-ray images were acquired. Mentioning that the knee joint position was tracked with skin markers is also necessary.

The last statement needs to be removed or clarified, as the high variability is between tasks and not necessarily between trials. Between trials implies the same participant completed the task differently, which the authors did not necessarily show.

7. PLOS authors have the option to publish the peer review history of their article (what does this mean?). If published, this will include your full peer review and any attached files.

Reviewer #2: No

---

## [Author Response · Author response to Decision Letter 2]

20 Jun 2025

Responses to Reviewer 2

We thank the reviewer for his valuable input and constructive feedback. Answers are below in a point-to-point manner in red. Changes made to the manuscript are highlighted in yellow, the page and line numbers mentioned refer to the adapted manuscript with tracked changes (Revised Manuscript with Track Changes).

Reviewer #2: The authors present a first-of-its-kind dual fluoroscope setup for tracking the entire gait cycle during walking and stair ascent/descent. The experimental advance is noteworthy, and the authors are commended for developing an impressive setup for tracking knee joint motion in healthy participants and later in patients following total knee arthroplasty (TKA). The manuscript provides sufficient detail to describe the system setup and, if others had the resources, replicate the work. (Of particular note is the description for the tracking of knee joint position and controller.)

On the other hand, the potential advance of this setup is clouded by the lack of gold standard measurements to compare tracking results against. Knee position tracking is an understandable target, but the authors’ setup may perform worse than single plane system or static dual fluoroscopy systems due to movement of the emitters and intensifiers.

Furthermore, the knee centre location was not observed on fluoroscope images, further shadowing the potential advancement described in the manuscript. For example, knee joint positions on the trailing edge of the fluoroscopes will have very little bone in the image for registration, but knee joint positions towards the leading edge will have much of the bone available for image registration.

We agree with the reviewer that direct validation against fluoroscopic images would strengthen confidence in the tracking setup. However, image acquisition was not feasible for ethical reasons during this proof-of-concept study, we explicitly clarified in the text (Page 9, Lines 311-312). This study was intended as a preliminary validation of the mechanical and tracking performance before future X-ray-based validation.

Another concern is the simulated cones for determining the fluoroscopes’ overlap. While the authors are commended for performing this calculation, the edges of the image intensifiers do not always yield usable image data for image registration. An object with the simulated cones dimensions, or close to it, imaged with both fluoroscopes, even statically, would lend confidence to the authors’ calculations. In the manuscript, all references to staying in the fluoroscopes’ FOV needs to include that the FOV was simulated.

We have now consistently clarified throughout the manuscript (abstract, methods, results, and conclusion) that the FOVs were simulated and that no X-ray images were captured during the study.

Abstract

Page 2, Line 52

Page 5, Line 138

Page 6, Lines 208, 211, 212, 220

Page 7, Lines 238, 240, 243

Page 8, Fig 5, Table 2

Page 10, Line 334

Also of concern, but less so, the references to ‘all gait speeds’ refers to a narrow, self-selected band of speeds in healthy participants. Patients following TKA may walk at slower speeds and, more importantly, with higher variability, which, again, brings into question the advancement described by the authors. While the authors acknowledged the lowest gait speed for a single trial, separating the trials into different groups would have made their point about variable speeds much stronger.

We acknowledge the reviewer’s concern and agree that our participant cohort represents a limited speed spectrum compared to clinical populations. However, we would like to clarify that our reference to “all gait speeds” is clearly contextualized within the manuscript as self-selected speeds by healthy participants. Our intent with this phrase is to demonstrate that the system places no constraints on participant speed during level walking, ramp climbing, and stair climbing, unlike previous moving fluoroscopes that were mechanically limited to slow gait.

While we did not group trials by speed for sub-analysis, the observed range — from below 1 m/s to well above average walking speed — encompasses the typical spectrum seen in both healthy and post-TKA individuals, as cited in our discussion. The key advancement, in our view, lies not in the specific speeds tested, but in the technical capability of the system to track across this full range without pre-programmed trajectories or speed limitations.

We believe the current phrasing and data presentation sufficiently support this point, and no changes to the manuscript were made in this regard.

Please change all references to ‘tracking’ with ‘knee joint position tracking’, because in the field of fluoroscopy tracking often refers to model-based tracking. And accuracy of tracking typically refers to comparisons to a gold standard for joint translations and rotations.

We appreciate the reviewer’s concern and fully agree that in the context of fluoroscopy, "tracking" can sometimes refer specifically to model-based tracking. However, in this manuscript, the term “tracking” is consistently and clearly used as referring to the real-time movement of the imaging system based on optical position estimates of the knee joint. This is a central and unambiguous concept throughout the paper.

To prevent potential ambiguity, we have now clarified this distinction in the Introduction.

Page 2, Line 25

“…joint position…”

We believe this clarification makes the intended meaning explicit, while avoiding the stylistic burden and redundancy of replacing all instances of “tracking” with “knee joint position tracking.” Therefore, the original terminology has otherwise been retained.

Specific Comments

Abstract

As referenced in the general comments, please clarify that the FOV was simulated, and that no images were acquired.

Abstract

“(no X-ray images were captured in this study)”

Introduction

No comments. Well written.

Methods

For lines 82-85, “As the radius of the intensifier sensor is 158mm, this setup (Fig 2) results in a capture volume with a length of 305mm in horizontal direction and a length of 445mm in lateral direction, and a length of 248mm in vertical direction at the intersection of the beam cone symmetry axes …”, were these theoretical values? Or experimental measures? Oftentimes the image intensifiers do not have usable image data all the way to the edge. Either way, the authors are commended for performing and reporting these calculations/measurements.

Page 3, Line 83-84

“(experimentally measured)”

On line 87, “The knee centre is tracked using an active marker tracking system (Fig 3):….” Was the active marker placed at a standard distance up the thigh? And did the tracking include an directionality to determine if the knee was bent or straight?

Page 3, Lines 91-92

”… at a comfortable and secure location on the thigh. Directionality of knee flexion/extension was not included in the tracking.”

The ‘Optical knee tracking’ section is noteworthy for the details provided on the experimental implementation. The text describes potential ‘rejected’ positions. Do the authors have an estimate for how often this occurred?

Typically below 10 percent.

Three to four hours for acquisition seems rather long if no fluoroscope images were acquired. As alluded to the in the last comment, did the authors experience setbacks with the system that would not be expected in a traditional motion capture setup?

The total duration included participant preparation, marker placement, and calibration. Some minor setup adjustments were needed that would not be required in conventional motion capture setups. These adjustments typically took up to 30 minutes.

Results & Discussion

Could the authors describe the results in Figure 5 in more detail? The text merely states that the tracking varied substantially between subjects. Also of note in Figure 5, it’s not clear what the y axis is plotting from the caption. Distance from the projected knee joint position to the center of the image intensifier?

Page 8, Fig 5

“…to the FOV centres…”

We respectfully note that the main finding illustrated by Figure 5 — substantial inter-subject variation in tracking performance within each task — is already clearly described in the manuscript. The purpose of this figure is to highlight that, despite similar task dynamics across participants, the projected knee centre positions vary between individuals, supporting our conclusion that geometric factors (e.g., lateral offset) influence tracking accuracy more than kinematic variability. We believe the level of detail currently provided appropriately reflects the intended message of this figure.

Lines 278-279, “leading to the negative values in the vertical plots of Fig 5.” The values in the plots look all positive. Are the authors referring to the positions plotted on the image intensifiers? Also, why did this not lead to the knee joint center position being outside the FOV, if the knee centre was lower the safety distance?

Page 8, Line 282

This was a typo. Corrected to refer to Fig 8, not Fig 5.

The vertical axis was constrained by a safety limit to avoid collision with floor fixtures. The knee could remain within the simulated FOV even at this lower bound.

Could the authors explain this statement in greater detail? Lines 307-308, “This study is limited by the fact that the X-ray system was not used due to ethical considerations.” Isn’t the point of the system to use x-rays? It would be understandable to assess tracking of the knee joint position to confirm performance prior to exposing participants to x-rays. But it’s not clear if that was what the authors are implying.

Page 9, Lines 311-312

“… as the purpose of this study was to confirm tracking performance prior to exposing participants to X-rays.”

Figures and Tables

Figure 1 is a nice zoomed-out view of the system, but, being the first figure, labels for the different components would benefit the readers who are not as familiar with fluoroscope systems.

Labels added.

The authors are applauded for the impressive polar plots in Figure 6 that clearly demonstrated the results.

For Figures 5, 6, 8, 10, and 11, it would help the reader interpret the images to know which direction the participants were moving in. This was mentioned in the text but not in the captions nor figures.

Added to captions of Fig 5, 6, 8, 10, and 11:

“For level walking, stair ascent, and ramp ascent, the subjects were moving from left to right, while for stair descent and ramp descent, the subjects were walking from right to left.”

Conclusions

Same as the abstract, please insert ‘simulated’ prior to FOV, and clarify that no x-ray images were acquired. Mentioning that the knee joint position was tracked with skin markers is also necessary.

Page 10, Line 335

“… using skin marker measurements of the knee joint position.”

The last statement needs to be removed or clarified, as the high variability is between tasks and not necessarily between trials. Between trials implies the same participant completed the task differently, which the authors did not necessarily show.

We appreciate the reviewer’s observation and agree that the current dataset was not designed to explicitly quantify trial-to-trial variability. However, we respectfully maintain the phrasing, as the key point being made refers not to observed variability in this cohort, but to the system’s design principle: our tracking method operates independently for each trial and does not require prior recordings or reference trajectories. This technical approach is specifically intended to handle potential variability between trials, which is known to be a relevant factor especially in clinical populations (e.g., TKA or neurological conditions), even if not directly measured in this study.

To clarify this distinction, we have revised the conclusion to better reflect that our statement refers to the capability and robustness of the system, rather than a finding demonstrated in the presented results:

Page 10, Lines 340-344

“By operating independently on each trial — without relying on prior trial data — our system is inherently suited to accommodate inter-trial variability, which is especially relevant in clinical contexts where such variability is common. While not directly assessed in this study, this robustness is a core feature of the tracking capability and supports broader applicability in patient populations.”

---

## [Editor Report · Decision Letter 2]

1 Jul 2025

A dual-plane fluoroscope to track joint kinematics during dynamic daily activities

PONE-D-24-27653R2

Dear Dr. Planta,

We’re pleased to inform you that your manuscript has been judged scientifically suitable for publication and will be formally accepted for publication once it meets all outstanding technical requirements.

Kind regards,

Aliah Faisal Shaheen

Academic Editor

PLOS ONE